# A Planar-Type Micro-Biopsy Tool for a Capsule-Type Endoscope Using a One-Step Nickel Electroplating Process

**DOI:** 10.3390/mi14101900

**Published:** 2023-10-04

**Authors:** Sangjun Moon

**Affiliations:** 1Department of Mechanical Convergence Engineering, Gyeongsang National University, Changwon 51391, Gyeongsangnam-do, Republic of Korea; nanobiomems@gnu.ac.kr; Tel.: +82-55-250-7304; Fax: +82-55-250-7399; 2Cyberneticsimagingsystems Co., Ltd., Changwon 51391, Gyeongsangnam-do, Republic of Korea; 3Department of Mechanical Engineering, Ulsan National Institute of Science and Technology (UNIST), Ulsan 44919, Republic of Korea

**Keywords:** planar-type micro-biopsy tool, one-step electroplating, capsule-type endoscope

## Abstract

Millimeter-scale biopsy tools combined with an endoscope instrument have been widely used for minimal invasive surgery and medical diagnosis. Recently, a capsule-type endoscope was developed, which requires micromachining to fabricate micro-scale biopsy tools that have a sharp tip and other complex features, e.g., nanometer-scale end-tip sharpness and a complex scalpel design. However, conventional machining approaches are not cost-effective for mass production and cannot fabricate the micrometer-scale features needed for biopsy tools. Here, we demonstrate an electroplated nickel micro-biopsy tool which features a planar shape and is suitable to be equipped with a capsule-type endoscope. Planar-type micro-biopsy tools are designed, fabricated, and evaluated through in vitro tissue dissection experiments. Various micro-biopsy tools with a long shaft and sharp tip can be easily fabricated using a thick photoresist (SU8) mold via a simple one-step lithography and nickel electroplating process. The characteristics of various micro-biopsy tool design features, including a tip taper angle, different tool geometries, and a cutting scalpel, are evaluated for efficient tissue extraction from mice intestine. These fabricated biopsy tools have shown appropriate strength and sharpness with a sufficient amount of tissue extraction for clinical applications, e.g., cancer tissue biopsy. These micro-scale biopsy tools could be easily integrated with a capsule-type endoscope and conventional forceps.

## 1. Introduction

Soft tissue inspection and sampling has been conducted using both an endoscope and an integrated dissection tool, which are used for abnormal tissue monitoring and biopsy, respectively [1,2,3,4]. In the case of gastrointestinal endoscopic monitoring, conventional flexible endoscopes have been used, but they frequently cause discomfort to patients during inspection and operation [5,6]. Recent advances in technologies related to medical robotics have led to the development of a capsular endoscope to monitor soft tissue and transfer images of the gastro-intestines with no discomfort to patients [7,8,9].

### 1.1. Endoscope Function

Despite the benefits that capsular endoscopes have over traditional endoscopes, their functionality is limited to solely transmitting images [10,11,12,13]. There are many candidate functions proposed to increase the utility of capsular endoscopes [14,15,16], such as biopsy [17,18,19,20], locomotion [21,22,23,24], and drug delivery [25,26,27]. For both minimal invasive surgery (MIS) and medical diagnosis, a millimeter-scale biopsy tool should be combined with an endoscope instrument to accomplish these aims [28,29]. Conventional biopsy tools have been made using mechanical tooling and successive assembly procedures for the complex biopsy structure [30,31]. However, the recently developed capsule-type small-form-factor endoscope has an easy and cheap fabrication process for small-size biopsy tools [32,33].

### 1.2. Endoscope Biopsy Tool Fabrication

Nonetheless, traditional machining techniques fall short when it comes to micro-scale instrument features. These limitations are twofold: first, the high costs associated with mass production, and second, the machining resolution, which extends only to tens of micrometers. Micro-machining technology is required to fabricate micro-scale biopsy tools that have both a sharp tip and other complex features, e.g., nanometer-scale end-tip sharpness and a complex scalpel design. Both a commercialized capsule-type endoscope and conventional mechanical forceps, as shown in Figure 1a, are used for intestine inspection and biopsy, respectively [29,34]. The conceptual schematic drawing of a planar-type micro-biopsy tool shows how the tools could be equipped with a conventional capsule-type endoscope to extract suspicious tissues for further in vitro examination; see Figure 1b.

As the size of micro-biopsy tools reaches the sub-mm scale, fabrication via conventional machining approaches becomes difficult [35,36,37]. Several fabrication processes based on silicon [38] and polymer materials [39] have been reported for micro-biopsy tools [40]. Among these fabrication processes, inductively coupled plasma (ICP) and wet etching, deep UV lithography, and a micro-molding process have been introduced for the fabrication of a long and slender biopsy tool, which could be integrated with an endoscope using a monolithic fabrication method. Compared to polymer materials, silicon has a high elastic modulus but shows brittle behavior, which makes it possible to leave sharp particles behind in the event of a fracture inside the target tissue. The polymer-based fabrication approach requires injection molding or an embossing process for mass production. It is also limited to a simple shank shape, which makes it difficult to form sharp scalpel tip features.

Here, we suggest and demonstrate a novel method for fabricating a complex-feature planar-type micro-biopsy tool using a simple one-step nickel electroplating process that could be cost-effectively adopted for mass production. By using a planar-patterned SU8 micro-mold and a one-step nickel electroplating process, robust and sharp biopsy tools can be easily fabricated using a monolithic process. A thick photoresist (SU8) mold created through a one-step simple lithography and nickel electroplating process can be adopted into various nickel deposition processes, such as electroforming [41,42], electroless plating [43], and electroplating techniques [44]. Electroplating leads to a greater than 99% nickel composition, which allows for thickness control during the deposition process. Based on these advantages, electroplating is the preferred nickel deposition technique for micro-scale feature formation since the deposition quality depends on the current forced through the electrolyte. Owing to its advantages, a nickel electroplating process was developed for a metallic, 99%-nickel, planar-type micro-biopsy tool with a long shaft and a sharp tip.

### 1.3. Content Summary

This study focused on designing, fabricating, and testing a metallic micro-scale biopsy tool that can be easily integrated with both capsule-type endoscopes and conventional forceps systems. These biopsy tools are thin, planar, and can fit into a small slot-type pocket in capsule endoscopes or be attached to conventional forceps via folding and laser spot welding. This fabrication method is simpler and more cost-effective than traditional techniques. Key design parameters like tip taper angles, tool geometries, and the number of cutting knives have been optimized for effective tissue extraction in clinical applications. Experimental results affirm the tool’s robustness and reliability across different design features.

## 2. Materials and Methods

### 2.1. Planar-Type Micro-Biopsy Design and Analysis

The failure modes of micro-biopsy tools include bucking by lateral force during the dissection process and bending by vertical force, which is experienced if the center line of the shank is not perpendicular to the surface of a target tissue. Considering the two failure modes and penetration force, a micro-biopsy tool was designed with four different shapes, with varying side scalpel types, shank diameters, and lengths. The design rules for the slender shank scalpel are followed by the previously described analytical method [45,46]. 

Two key methods are utilized for assessing the failure modes of a micro-biopsy tool. The first method investigates buckling, considering factors such as Young’s modulus and the yield strength of the material. For a tool made of nickel, susceptibility to buckling is significant. The second method focuses on calculating the maximum transverse tip force, using variables like the distance from the neutral axis to the shank’s outermost edge and the second moment of the cross-sectional area. In the case of a tool made of nickel with specific dimensions and material properties, such as a 3 mm long shank length and a 67.0 µm radius of gyrus, the tool is prone to buckling. Meanwhile, following the assumption of linear elastic deformation, the maximum force exerted by the 3 mm long micro-biopsy tool is calculated to be 0.27 N, equivalent to a bending moment of 0.81 mN-m.

However, the analytical approach cannot be adopted for a complex cross-sectional area, as shown in Figure 2a. For a complex shank shape and the cross-sectional area, the maximum stress can be figured out with a numerical approach. Stress analysis of the designed micro-biopsy structure was performed using a commercial simulation tool, COMSOL Multiphysics^®^ modeling software, V5.0 which tested three different magnitudes of vertical force at the tip of the micro-biopsy tool. Using the mechanical properties of nickel and material library data, maximum Von Mises stress was found at the end of the biopsy tip when lateral force was applied and at the connection between the shank and base when vertical force was applied.

### 2.2. Planar-Type Micro-Biopsy Fabrication

The fabrication process of the micro-biopsy tool was composed of two main parts, SU8 (SU-8 100, MicroChem, Seoul, Republic of Korea) molding and a nickel electroplating process, as depicted in Figure 3. One is a thick photoresist patterning process using a commercial product, SU8, for nickel electroplating molding, and the other is a nickel electroplating process guided by the previously fabricated SU8 mold pattern. 

The mold fabrication process was as follows: (i) A 4″ diameter and 500 µm thick test-grade silicon wafer (Fine Chemical Industry, Republic of Korea) was cleaned in alkaline detergent, rinsed in distilled water, and blow-dried with nitrogen. Then, 30 nm thick titanium (Ti) was sputtered on the cleaned silicon substrate with a cluster sputter (SRN-110, Sorona, Republic of Korea) as an electroplating seed layer. 

(ii) Negative photoresist SU8 was spread onto the metallized glass substrate via spin coating at 500 rpm for 15 s and 2000 rpm for 40 s to obtain a 150 µm thick photoresist layer. Then, the spin-coated wafer was baked at 65 °C for 6 min and 95 °C for 50 min in a convection oven. The soft-baked wafer was overlaid with a chromium mask at the mask aligner (Suss Microtec MA8-GEN3 mask aligner, German) and exposed to UV light for 17 s at 250 mJ/cm^2^ followed by post-exposure baking at 65 °C for 6 min and 95 °C for 20 min in a convection oven. The hard-baked SU-8 was developed in a SU-8 developer for 15 min, rinsed with de-ionized (DI) water, and blow-dried with nitrogen gas. 

(iii) Based on the SU8 micro-mold structure, nickel was plated via the following procedures. The nickel electroplating solution was a nickel sulfamate electrolytic solution buffered with boric acids (450 mL per 1 L of DI water of nickel sulfamate (Ni(SO_3_NH_2_)_2_), 37.5 g of boric acid (H_3_BO_3_), and 3 g of sodium dodecyl sulfate or lauryl sulfate (C_12_H_25_NaO_4_S), pH 3.2, 55 °C), as recommended by the chemical supplier (SungWon Forming, Republic of Korea). The electroplating bath was controlled to obtain reliable and repeatable electroplating quality. The electroplating conditions were monitored and maintained using commercial electroplating equipment (SWPMC-L01-P01, SungWon Forming, Republic of Korea) which generated stable unipolar rectangular pulses through a closed-loop current controller. During the plating process, the magnetic stirrer served to maintain an even ion distribution throughout the whole bath, which improved the uniformity of plating thickness. Using a dummy Si wafer, the operational conditions were investigated for the current amplitude, the duty cycle, and the pulse time. The electroplating bath was heated to 55 °C, and a maximum current density of 10 mA/cm^2^ (for slow, stress-free plating 3 mA/cm^2^) was used to obtain a stress-free stand-alone microstructure without a holding substrate. 

(iv) After nickel electroplating with the mold, the SU8 mold was swelled out using a PG remover. Some scum on the SU8 mold remained at the shallow region of the nickel structure after the swelling process. On a silicon wafer, titanium was evaporated to obtain a seed layer during the electroplating process and a release layer during the mold removal process. 

(v) Using isotropic dry etching with SF_6_ and O_2_ plasma at 200 W for 40 min (FabStar-ICP, TTL, Ireland), all the scum was successfully removed. The process requires only a single mask, making it scalable, since it depends on the lithography equipment yield for commercialization of the tool. The fabricated biopsy tool can be mounted to an SUS wire (stainless-steel wire) and a capsule-type endoscope mock-up model.

### 2.3. Fabrication Summary

This paper details a fabrication procedure for a planar micro-biopsy tool involving the use of thick photoresist SU8 and nickel electroplating. The multi-step process, illustrated in Figure 3a, starts with a Ti seed layer for electroplating and uses SU8 as a mold for nickel plating. The mold is then chemically removed, and any residues are eliminated through dry etching. This method is optimal for mass production as it relies on a single mask and can be adapted for batch electroplating. Importantly, Figure 3b highlights that the intricate design features of the tool are maintained throughout the process. Chemical etching selectively targets the polymer mold, ensuring the final metal product is free from debris that could irritate tissue in biopsy applications.

### 2.4. Mouse Intestinal Tissue Staining

A male FVB/N (FVB) mouse, at the age of 6 weeks, was sacrificed under carbon dioxide euthanasia using a transparent plastic chamber filled with CO_2_. The small intestine was exposed via surgical operation for the in vitro tissue dissection experiment. A 20 mm × 20 mm skin patch was removed from the inner side of the abdominal region for the dye diffusion test. First, half of the shank length, 1.5 mm, from the scalpel tip was immersed into a staining solution, bromophenol blue (3′,3″,5′,5″-Tetra bromophenol sulfonphthalein, Sigma-Aldrich, Republic of Korea). 

The staining solution filled the surface and pockets through the grooves of the fabricated micro-biopsy tools via hydrophilic capillary force. The bromophenol blue-containing micro-biopsy tool was manually pushed with forceps into the mouse skin patch. Agarose gel, which is composed of agarose powder (Gibco BRL, life technologies, Republic of Korea) and 2% distilled water, was used for the dye diffusion test. Next, the open tissue surface of the small intestine region was pricked with the fabricated micro-biopsy tools to extract the mouse intestine tissue. After being pulled out from the intestine tissue, the fabricated micro-biopsy tools were stained and de-stained, as previously described [29]. In brief, all biopsy tools were immersed in a protein-staining solution, which was composed of 45% methanol, 10% glacial acetic acid, and 0.25% CBB G250 (Coomassie Brilliant Blue G250, life technologies, Republic of Korea) in distilled water. After 30 min at room temperature, the excess dye was removed through several applications of staining solution without CBB G250. The existence of any stained protein on the biopsy tools was examined through a bright-field optical microscope.

## 3. Results and Discussion

### Fabrication Results

The design parameters for the micro-biopsy tool were established based on several key functionalities: the mechanical strength needed to withstand bending forces during tissue penetration, the sharpness of the tip for effective tissue dissection, and the capacity of the tissue container to hold an adequate volume for in vitro diagnosis. Following these criteria, four different micro-biopsy tools with different side scalpel types, shank diameters, and lengths were tested, as shown in Figure 2a. The shank length needed to extract tissue according to the intestine wall thickness was determined to be 3 mm, which is sufficiently long to reach deep tissue regions, i.e., fascia. The scalpel outline and its dimensions were classified into four different shapes, as shown in the schematic table. There were two solid shank types and two hollow shank types, which are based on both the mechanical strength of the biopsy tool and extraction volume of tissue, respectively. The inner and outer diameters ranged from 100 to 400 µm considering the minimum feature sizes needed for a nickel electroplating process. The number of scalpels was 4 and 10 each, which had 57 and 75 µm shank rib thicknesses. The tool base was designed as a 3 × 5 mm square to be compatible with the assembly process of a capsule-type endoscope and forceps. Finally, four types of micro-biopsy features were chosen and examined for further mechanical analysis and in vitro dissection experiments.

Stress evaluations of the micro-biopsy apparatus were carried out via COMSOL Multiphysics^®^ simulations, as depicted in Figure 1. This simulation validated the upper tolerances for bending and buckling stress in the instrument. The design of the tool was tailored to mitigate penetration and dissection forces, eliminating the risks of shank buckling and tip fractures. For forces below 0.3 N, no buckling phenomena were observed (Figure 2b(ii)). At a lateral force of 0.3 N, a peak Von Mises stress of 3.36 × 10^7^ N/m^2^ was identified at the tip. Bending stress reached a maximum at the interface between the shank and the base, registering 3.49 × 10^7^ N/m^2^ and 1.05 × 10^8^ N/m^2^ for vertical forces of 10.0 mN and 30.0 mN, respectively (Figure 2b(iii,iv)).

Analytical calculations based on linear elastic deformation models suggested that the 3 mm long tool can withstand a maximum force of 0.27 N, translating to a bending moment of 0.81 mN-m, as illustrated in Figure 3. These data align with the simulation findings, verifying the tool’s resilience to bending and buckling stress. Additional results concerning alternative designs can be found in Appendix A. Considering the material’s higher stiffness compared to tissue, applied forces are likely to stay under the maximum stress limit of 510 MPa, especially when the tip is sharpened to a thinness of 150 µm, allowing for safe tissue penetration without tip fractures.

The stress tests and deformation models offer significant evidence for the mechanical durability of the micro-biopsy device against forces that could cause it to bend or buckle. Both the COMSOL Multiphysics^®^ simulations and the analytical estimates confirm the device’s ability to perform as expected. However, this study has limitations in accounting for variations in material properties like Young’s modulus, homogeneity, and heterogeneity, specifically when comparing traditional nickel alloys to electroplated nickel. Additional studies are recommended for a more comprehensive understanding of these material differences.

After the fabrication of the micro-biopsy tools, as shown in Figure 4, the various shapes of micro-tools could be integrated with a conventional capsule-type endoscope and forceps as dissection tools with a planar-type thin micro-scale form factor. Four different free-standing planar-type biopsy tools were prepared for further experimentation through a reactive ion etching de-scumming process (RIE), and released from a titanium (Ti)-coated silicon substrate for the electroplating process. Four shanks with the same scalpel and geometries work with one set of micro-biopsy tools, which were fabricated with an assembly base structure to serve as an integrated component, as shown in Figure 4a.

The planar-type micro-biopsy tools were integrated with a capsule-type endoscope mock-up product (a rapid prototyping part made with a 3D printer) which was inserted into an opening slot equipped with spring-loaded ejection module, as seen in Figure 4b(i), and integrated with a conventional forceps tool via laser spot welding, as seen Figure 4b(ii). The planar-type design of the micro-biopsy tools makes them suitable to be assembled with a capsule-type endoscope with a small form factor, which requires only a thin, narrow slot less than 200 µm in size. Moreover, a thin metal basement for the four shanks can be rolled up and assembled with a laser spot welding process for the conventional forceps. These features demonstrate the applicability of the fabricated micro-biopsy tool for conventional microsurgery.

Dye diffusion and in vitro dissection experiments were performed with the planar-type micro-biopsy tools, as shown in Figure 5. The fabricated micro-biopsy tools were immersed in a staining solution (bromophenol blue) to allow the liquid to fill the hollow spaces of the shank and the scalpel surface using the hydrophilic characteristic of the metal surface and a manual tool holder, as shown in Figure 5a(i). The mouse skin and agarose gel were used as phantom intestine tissue. The staining solution on the surface and in the container diffused through the phantom tissue just after the micro-biopsy tool was pushed in. The blue-colored dye diffused through the shank grooves on the micro-biopsy tools after administration on the mouse skin, as seen in Figure 5a(ii), and the lump of agarose gel, shown in Figure 5a(iii). The diffused region covered different areas based on the material characteristics of the phantom tissue used. The four shanks of the micro-biopsy tool showed a line of the diffused pattern of the staining solution when applied to the agarose phantom tissue, but not the mouse skin phantom, which can be used to determine drug administration sites. The shanks of the micro-biopsy tool can easily penetrate the cell and tissue without any deflection or fracturing. The pocket volume of the shank allows it to hold more transferable liquid, but means it has less structural strength during the insertion and extraction process. The staining solution can be substituted with a target drug for administration into a suspicious spot for direct drug delivery. 

The in vitro dissection experiment with surgically opened mouse intestine tissue is shown in Figure 5b. For the in vitro micro-biopsy function test for an intestine tissue dissection, the small intestine of a male FVB/N (FVB) mouse was exposed via surgical operation. The surface of the small intestine was pricked with the biopsy tool, pulled out, and cut from the normal tissue, as shown in Figure 5b(i). After being pulled out, the whole biopsy tool, which contained the dissected intestine tissue, was immersed in a protein-staining solution following the method described earlier, as shown in Figure 5b(ii). The existence of any stained tissues on the biopsy tool was examined through a bright-field optical microscope. The four different types of shank and scalpel shapes suggested for the micro-biopsy tool, type 1 to type 4, are shown in Figure 5b(iii). Types 1 and 2 do not have any hollow shank which can contain dissected tissue, while types 3 and 4 do have hollow shanks. The solid shank types, types 1 and 2, however, can hold a part of the dissected tissue at the outer region of the shank, as shown in Figure 5b(iii). The sharper and thinner scalpel shape, type 2, can hold more tissue at the peripheral of the shank, allowing it to cut off more tissue. Among the tools with the same shank diameters and number of scalpels, the hollow-type micro-biopsy tools, types 2, 3, and 4, extracted more tissue than the solid shank types 1 and 2. According to the various design parameters, type 3 and type 4, which have a hollow shank pocket, showed that more tissue can be extracted through the pocket, but minor shank bending and deformation were observed when these were pulled out; see the arrow markers as shown in Figure 5b(iii). Some tested hollow-type micro-biopsy tools bent, but did not break or leave behind debris. This indicates that the fabricated biopsy tool can extract sufficient tissue for a biopsy, but the strength of the micro-biopsy tool should be optimized according to the target tissue and applications.

## 4. Discussion and Conclusions

The electroplated micro-biopsy tool showed sufficient mechanical strength to penetrate a mouse intestine even though its shank length was over 3mm long and it had a hollow shank pocket for tissue collection. The nickel electroplating process molded using a thick photoresist was used to fabricate micro-biopsy tools with various tip shapes and designs. The specific mechanical properties can vary greatly depending on the conditions of deposition, post-treatment, and even the quality of the base material [47]. Recently, simulation methods have also been used to predict mechanical characteristics through experiments on the conditions of deposition [48] and simulations on fluid and electric field distribution [49]. Through careful control of deposition conditions and by employing advanced simulation techniques, it is possible to tailor the mechanical properties of electroplated nickel to meet these specific needs.

This process shows the metal-based micro-biopsy tools are robust and can be easily fabricated using one-mask lithography and a one-step electroplating process. The planar-type micro-biopsy tools have been suggested for integration with the capsule-type endoscope and conventional forceps system [50]. For more recent work based on another mechanism, see the studies investigating a magnetic torsion spring mechanism for a wireless biopsy capsule [51] and a magnetic capsule endoscope carrying, releasing, and retrieving untethered microgrippers [52], which are proposed to be equipped to the small capsule.

Micro-scale biopsy tools with a 3 mm long, 150 µm thick shank and a nickel-based composition were used to successfully extract sufficient mouse intestinal tissue by cutting the inner tissue layer. The scalpels were designed for the extraction of specific tissues from the target depths. Both computational simulations and laboratory experiments point to an unexpected finding that the area most susceptible to deformation in the biopsy tool is not the tip, which is often assumed to be the weak point, but rather the connection where the needle attaches to the base. This realization necessitates a shift in focus, directing attention to the structural integrity of the needle-base junction. Based on these findings, this study calls for a comprehensive redesign of the tool’s 2D schematic. Specifically, the inclusion of geometric features like rounded or chamfered shapes is recommended as a strategy to enhance the tool’s overall mechanical stability and durability.

However, the extracted inner tissues were mixed with outer tissue during their removal with the micro-biopsy tool since the inner tissue could not be protected during extraction. For further applications and depth profiling of the extracted tissue, an active dissection tool could be integrated with the micro-biopsy tool, which has been introduced with a PZT-based vibration mechanism [38]. At the base of the fabricated micro-biopsy tool, the active actuator, e.g., buzzer, could be assembled with a spin-coated epoxy glue. Without using a thick silicon substrate, the thin metal substrate could be integrated with an actuator with a proper dielectric polymer material between the metal and PZT layer, without diminishing the one-step mass production advantages. 

The surface of the fabricated micro-biopsy tool after the one-step nickel electroplating process did not need additional surface treatments such as a grinding and polishing process after being releasing from the silicon substrate with a dry RIE (reactive ion etching) process. However, when the micro-biopsy tool is designed and applied for either tissue extraction and drug delivery, it should be coated with a bio-inert polymer preventing tissue irritation and containing a target drug during administration. For the conformal coating of the tool surface without deterioration of its advantages using the one-step electroplating process and batch manufacturing, parylene-C could be a candidate polymer since the polymer is bio-inert and can be coated via the CVD (Chemical Vapor Deposition) process, which would cover the entire micro-biopsy tool surface. Following our previous research on CVD-based cross-linked polymers, parylene-C and parylene-H, as different forms of functional polymer networks, these results demonstrate good conformal coating sufficient to prevent chemical adsorption [53] and to utilize an anti-body linkage [54,55]. Based on these fabrication advantages and in vitro tissue dissection results, the micro-biopsy tool could be utilized for micro-surgery applications via integration with a conventional capsule-type endoscope with a bio-inert surface to minimize tissue irritation and provide drug delivery functionality.

Though the needle design and modeling were improved, the entire analysis was affected by a major issue. Unfortunately, if the needles are operated by hand, as demonstrated in the experimental part, the results will be extremely different with respect to those we could obtain by operating the needles with a capsule. In fact, without any anchoring mechanism of the capsule, the needle may not be able to penetrate through the tissue and the overall procedure could be compromised. In addition, the integration of the needle into an endoscope produces different results for both manual operation and capsule operation. Based on this consideration, a magnetically anchored capsule mechanism presents a promising approach to utilizing the current planar-type biopsy tool. Moreover, the same applies for simulation with COMSOL: the force application and distribution may be very different when the needles are integrated into a real capsule, as shown in Figure 4b, and when they must be pulled out and retrieved without any consideration of the anchoring force of the capsule.

In conclusion, we have demonstrated a micro-machined biopsy tool which was fabricated with a one-step nickel electroplating process. The electroplated nickel micro-structure features a planar type, having a two-dimensional shape so that it is suitable to be equipped with a capsule-type endoscope. The planar-type micro-biopsy tools were designed and fabricated following important design parameters, i.e., overall form factor, bending strength, sharpness of the tool tip, and pocket volume. Various micro-biopsy tools which have long shafts and sharp tips have been easily fabricated using a thick photoresist (SU8) mold via a one-step simple lithography and nickel electroplating process. The characteristics of various micro-biopsy tool design features include tip taper angle, tool geometries, and cutting scalpel number. The fabricated micro-biopsy tools demonstrated proper strength and sharpness and have been successfully optimized for extracting a sufficient amount of tissue from a mouse intestine. The in vitro tissue dissection experiment also showed that the tools were able to penetrate the tissue with sufficient protein dye delivery for clinical application, e.g., cancer tissue biopsy and drug administration. Moreover, the micro-biopsy tool was manufactured cost-effectively via the one-step electroplating process using micro-fabrication batch processing. We have brought the nickel micro-biopsy tool one step closer to compact and ergonomic ready-to-use micro-surgery applications. These micro-scale biopsy tools can be easily integrated with a capsule-type endoscope and conventional forceps.

## Figures and Tables

**Figure 1 micromachines-14-01900-f001:**
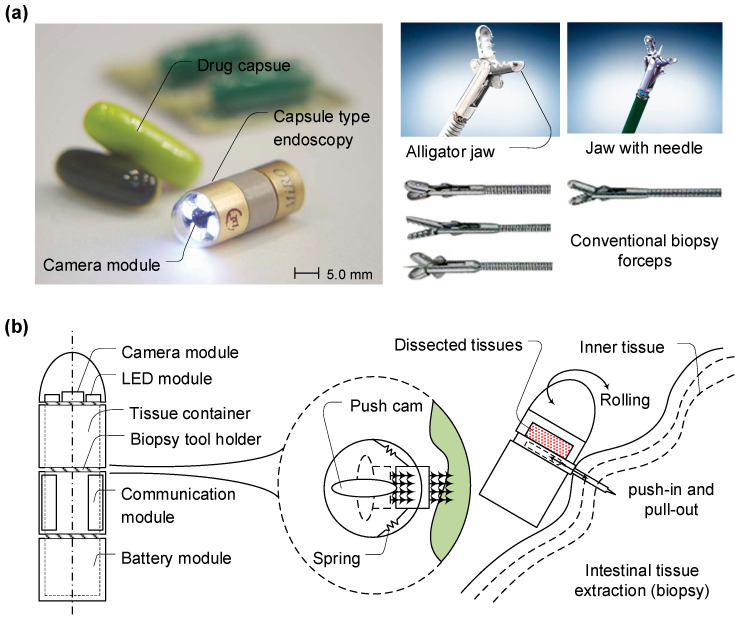
Conceptual schematic drawing of a planar-type micro-biopsy tool equipped with a conventional capsule-type endoscope which has the function of conventional mechanical forceps. (**a**) Pictures of both a commercialized capsule-type endoscope and conventional mechanical forceps used for intestine inspection and biopsy, respectively (MicroCAM, Olympus, Seoul, KR). (**b**) Concept of a planar-type micro-biopsy tool which can be integrated into a capsule-type endoscope to extract suspicious tissues for further in vitro examination during inspection of the inside surface of an intestine.

**Figure 2 micromachines-14-01900-f002:**
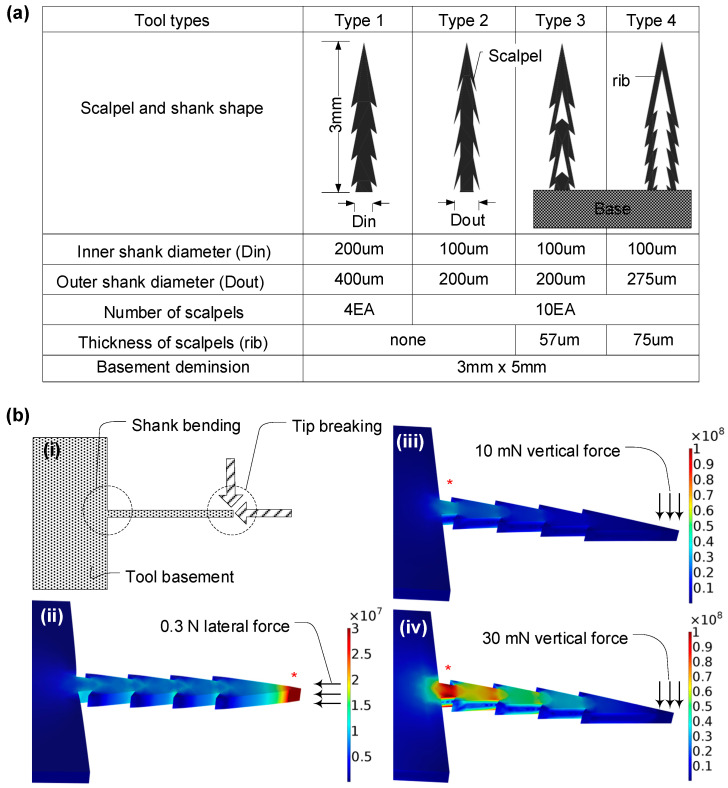
Design of a micro-biopsy tool with four different side scalpel shapes, shank diameters, and lengths. (**a**) Schematic table of designed dimensions for four different micromachined biopsy tools. The design rules are based on both the mechanical strength of the biopsy tool and extraction volume of tissue. (**b**) Stress analysis of the designed micro-biopsy structure using a commercial simulation tool, COMSOL Multiphysics^®^ modeling software, which tested three different magnitudes of vertical force at the tip of the micro-biopsy tool. Maximum Von Mises stress is shown at the end of the biopsy tip when lateral force is applied and at the connection between the shank and base when vertical force is applied.

**Figure 3 micromachines-14-01900-f003:**
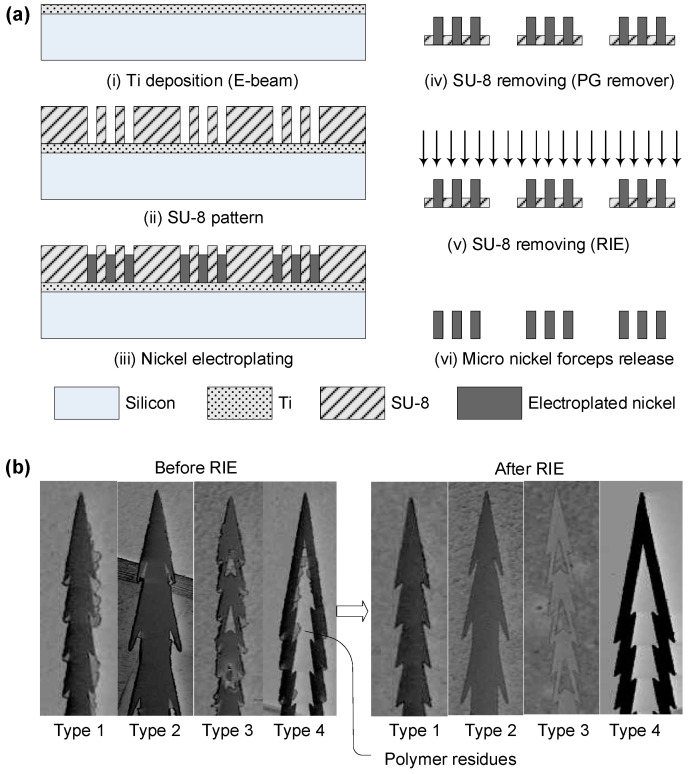
Fabrication flowchart of a planar-type micro-biopsy tool based on both a thick photoresist, SU8, and nickel electroplating process. (**a**) The fabrication process starts with deposition of electroplating seed layer with Ti (**i**), and the thick photoresist, SU8, is patterned to be used as a mold for the next electroplating step (**ii**). Based on the mold structure, nickel is plated (**iii**) and the mold is removed using a chemical etchant, PG remover (**iv**). Finally, the remaining residues on the nickel surface are removed via a dry etching process using RIE (**v**). The process only needs a single mask, making it a scalable process, since it depends on lithography equipment yield for commercialization of the tool. (**b**) De-scumming process to remove thick photoresist, leaving only metal parts after the dry etching process.

**Figure 4 micromachines-14-01900-f004:**
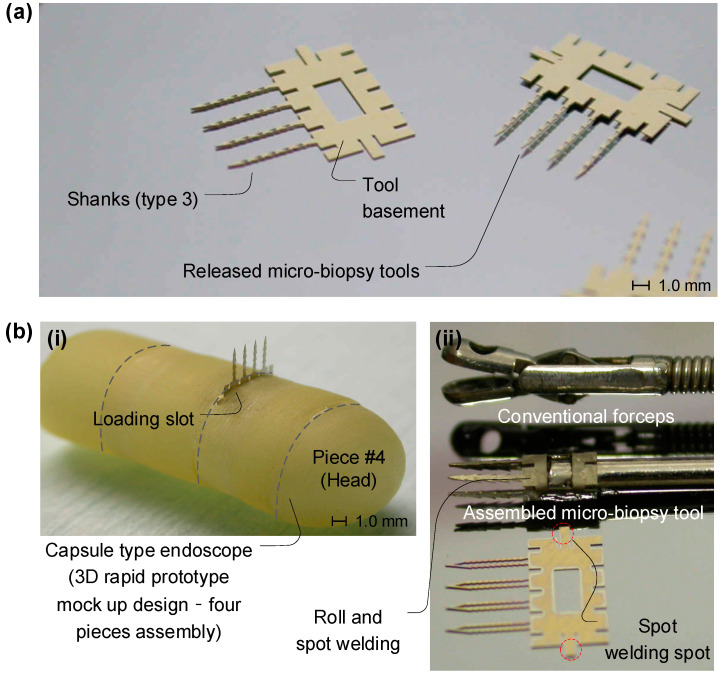
Integration of the micromachined planar-type biopsy tool. (**a**) Four different free-standing planar-type biopsy tools, which were released from an electroplating substrate and Ti-coated silicon substrate through an RIE de-scumming process. Four shanks work with one set of micro-biopsy tools, which were fabricated with an assembly base structure for part integration. (**b**) The planar-type micro-biopsy tools integrated with a capsule-type endoscope mock-up product (a rapid prototyping part made with a 3D printer), which was inserted into an opening slot equipped with a spring-loaded ejection module (**i**) and integrated with a conventional forceps tool via laser spot welding (**ii**).

**Figure 5 micromachines-14-01900-f005:**
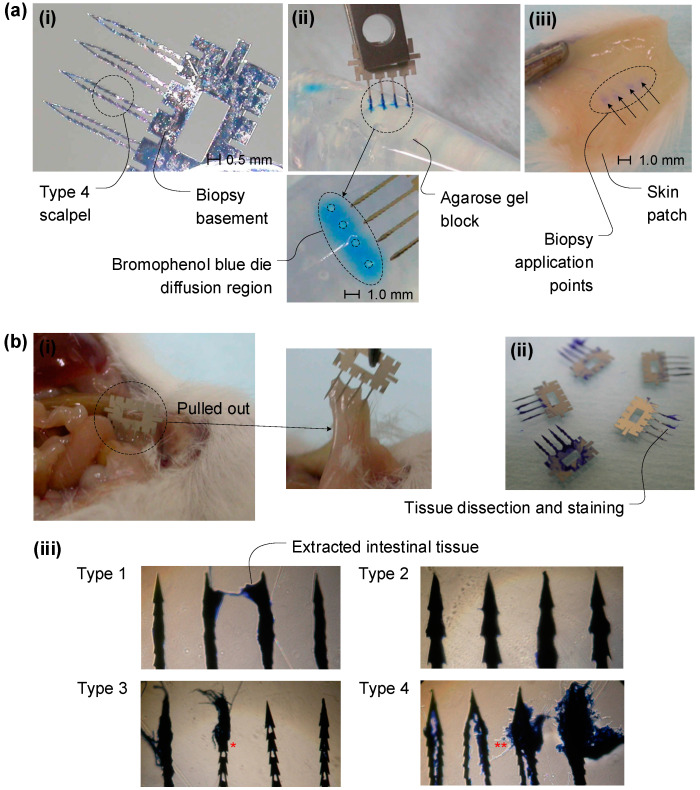
Dye diffusion and in vitro dissection experiment with the planar-type micro-biopsy tools. (**a**) Blue dye is diffused through the shank grooves on the biopsy tools, administrated to a lump of agarose gel (**ii**) and a piece of mouse skin (**iii**). The shanks of the micro-biopsy tool can easily penetrate the gel and tissue without any deflection or fracturing. The pocket volume of the shank allows it to hold more transferable liquid, but means it has less structural strength during the insertion and extraction process. (**b**) In vitro dissection experiment with surgically opened mouse intestine tissue. The tissue was extracted by manually pulling it in and out with the planar-type micro-biopsy tools. The stained tissue after extraction was blue under the optical microscope. Type 3 * and Type 4 **, which have more shank pocket volume, showed that more tissue can be extracted through the pocket, but minor shank bending and deformation were observed when it was pulled out.

## Data Availability

Not applicable.

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
