# Peer review of "A Planar-Type Micro-Biopsy Tool for a Capsule-Type Endoscope Using a One-Step Nickel Electroplating Process"

_micromachines, 2023, doi:10.3390/mi14101900_

Round 1

Reviewer 1 Report

In the submitted manuscript, the author adopted the electroplating technique to modify the tips of the micro-biopsy tool. The simulations and experiments demonstrated the proper strength and sharpness for tissue extraction in clinics. However, I still have several concerns as listed below.

1.     The tips are vertical to the capsule surface all the time. Then, how to make sure to extract tissues at the desired position, since the tips will penetrate the tissues at any position when they get contact with the digestive tract.

2.     Will the tips lead to perforation in the digestive tract? How to make sure the safety in clinical application?

3.     There are too many errors in the References. For example, the page numbers are wrong in Ref 1, 3, 6 etc. The journal name in Ref 10 should be in abbreviation. Please go through the Reference Part carefully and correct all the errors.

Author Response

Answers to the reviewer’s comments

On behalf of my co-authors, I am re-submitting the enclosed material after revision for possible publication in your journal. We sincerely thank the reviewers for their careful reading of the manuscript and valuable comments. We have made revisions according to the reviewers’ helpful comments and suggestions, as described below. The revised portions of the manuscript are highlighted in blue.

In the submitted manuscript, the author adopted the electroplating technique to modify the tips of the micro-biopsy tool. The simulations and experiments demonstrated the proper strength and sharpness for tissue extraction in clinics. However, I still have several concerns as listed below.

  1. The tips are vertical to the capsule surface all the time. Then, how to make sure to extract tissues at the desired position, since the tips will penetrate the tissues at any position when they get contact with the digestive tract. Will the tips lead to perforation in the digestive tract? How to make sure the safety in clinical application?

Ans.: A method for position control is necessary when the capsule is attached to the gastrointestinal tract using an external anchor. When attached to a specific location through an anchor, a tip protrudes from the outside of the capsule, penetrating the tissue due to an internal mechanism consisting of a spring-cam. The external protrusion length is determined by the protrusion mechanism of the spring-cam structure and the capsule's anchor attachment method. In clinical trials, the capsule includes an internal tip structure, making it a safe design. However, when the tip protrudes externally, the length must be designed to not violate the thickness of the gastrointestinal wall.

(Line : 400-412).

The needle design and modeling are appreciated but the entire analysis is affected by a major issue. Unfortunately, if the needles are operated by hand, as demonstrated in the experimental part, the results will be extremely different respect to what we can obtain by operating the needles onto a capsule. In fact, without any anchoring mechanism of the capsule, it can happen that the needle cannot penetrate through the tissue and the overall procedure could be compromised. In addition, the integration of the needle on board an endoscope produces different results from both manual operation and capsule operation. Based on this consideration, a magnetically anchored capsule mechanism will be a promising approach to utilize current planar type biopsy tool. Moreover, the same applies for the simulation by COMSOL: the force application and distribution can be very different when the needles will be integrated into a real capsule as in Fig. 4(b) and when they must be protruded and retrieved without any con-sideration of the anchoring force of the capsule.

  1. There are too many errors in the References. For example, the page numbers are wrong in Ref 1, 3, 6 etc. The journal name in Ref 10 should be in abbreviation. Please go through the Reference Part carefully and correct all the errors.

Ans.: Thank you very much for reviewing the error. All the references were modified following the reviewer’s comment. Additionally, we have changed the format of all other references by referring to the MDPI endnote style file.

Reviewer 2 Report

The paper describes the manufacturing and early in vitro experiments with a novel micro-scale biopsy design intended to be used with capsule-type endoscopes. The topic of the paper is relevant, the described results are mostly well-supported and important and the conclusions are sufficient as well, however, the reviewer has some remarks, questions and suggestions regarding the paper:

- Since the paper does not only describe 1 aspect of the biopsy too, but delves into manufacturing, simulation, different design options, and in vitro testing as well, a "content summary" in the introduction would highly improve readability (preferably along with the introduction of more subsections). - In the Result section description of methods can often be found which don't belong there, and they are often only repeating information already written in the Methods section. For example, the paragraph starting at line 252, and the one starting at 286. The reviewer suggests separating results from methodology and leaving only the description of results in the Results section (mainly in texts, but in figures as well). - The significance of the 2 equations is not clear. It is stated that they can not be used in this case, so why are they presented in the paper along with numbers, but without any computational results? It is suggested either to exclude the equations from the paper, or at least use them with a simplified model and show the difference between those results and the simulated results.  - The exact material-model is not given at the simulations. I.e., does the electroplating alter the material properties? How big a mistake is it to handle it as a homogenous/isotropic material? - No detailed conclusion is drawn from the various simulated scenarios: What limitations were found? where does the model need modifications? Where will it - probably - break?  - The paper would highly benefit from any kind of real mechanical measurement that would support the validity of the simulations. The authors shall do a more profound search into the state-of-the-art, and present that.

Numerous typos and grammatical errors can be identified.

Author Response

Answers to the reviewer’s comments

On behalf of my co-authors, I am re-submitting the enclosed material after revision for possible publication in your journal. We sincerely thank the reviewers for their careful reading of the manuscript and valuable comments. We have made revisions according to the reviewers’ helpful comments and suggestions, as described below. The revised portions of the manuscript are highlighted in blue.

The paper describes the manufacturing and early in vitro experiments with a novel micro-scale biopsy design intended to be used with capsule-type endoscopes. The topic of the paper is relevant, the described results are mostly well-supported and important, and the conclusions are sufficient as well, however, the reviewer has some remarks, questions, and suggestions regarding the paper:

  1. Since the paper does not only describe 1 aspect of the biopsy too, but delves into manufacturing, simulation, different design options, and in vitro testing as well, a "content summary" in the introduction would highly improve readability (preferably along with the introduction of more subsections).

Ans.: For improved readability, the introduction section of the research has been modified according to the suggestions made by the reviewer, and the content is as follows.

(Line : 36-38).

Endoscope function

Despite the benefits that capsular endoscopes have over traditional endoscopes, their functionality is limited to solely transmitting images

(Line : 46-50).

Endoscope biopsy tool fabrication

Nonetheless, traditional machining techniques fall short when it comes to micro-scale instrument features. These limitations are twofold: first, the high costs associated with mass production, and second, the machining resolution, which extends only to tens of micrometers

(Line : 91-100).

Content summary

This study focuses on designing, fabricating, and testing a metallic micro-scale biopsy tool that can be easily integrated with both capsule-type endoscopes and conventional forceps systems. These biopsy tools are thin, planar, and can fit into a small slot-type pocket in capsule endoscopes or be attached to conventional forceps via folding and laser spot welding. The fabrication method is simpler and more cost-effective than traditional techniques. Key design parameters like tip taper angles, tool geometries, and the number of cutting knives have been optimized for effective tissue extraction in clinical applications. Experimental results affirm the tool's robustness and reliability across different design features.

  1. In the Result section description of methods can often be found which don't belong there, and they are often only repeating information already written in the Methods section. For example, the paragraph starting at line 252, and the one starting at 286. The reviewer suggests separating results from methodology and leaving only the description of results in the Results section (mainly in texts, but in figures as well).

Ans.: The methods described in the Results section have been separated into a Methods section and a Results section. Specifically, the paragraph on lines 252-286 has been modified, and other paragraphs requiring separation have been rewritten as follows.

(Line : 192-201).

Fabrication summary

The study details a fabrication procedure for a planar micro-biopsy tool, involving the use of thick photoresist SU8 and nickel electroplating. The multi-step process, il-lustrated in Fig. 3(a), starts with a Ti seed layer for electroplating and uses SU8 as a mold for nickel plating. The mold is then chemically removed, and any residues are eliminated through dry etching. This method is optimal for mass production as it relies on a single mask and can be adapted for batch electroplating. Importantly, Fig. 3(b) highlights that the intricate design features of the tool are maintained throughout the process. Chemical etching selectively targets the polymer mold, ensuring the final met-al product is free from debris that could irritate tissue in biopsy applications.

(Line : 225-229).

3.1. Fabrication Results

The design parameters for the micro-biopsy tool were established based on several key functionalities: the mechanical strength needed to withstand bending forces during tissue penetration, the sharpness of the tip for effective tissue dissection, and the capacity of the tissue container to hold an adequate volume for in-vitro diagnosis.

  1. The significance of the 2 equations is not clear. It is stated that they cannot be used in this case, so why are they presented in the paper along with numbers, but without any computational results? It is suggested either to exclude the equations from the paper, or at least use them with a simplified model and show the difference between those results and the simulated results.

Ans.: The two equations presented in the paper were used for selecting parameters to design the initial model. Following the reviewer's suggestions, unclear equations were excluded, and the initial calculated values using a simplified model were compared with simulation values to present the differences. The changes are as follows.

(Line : 109-118).

Two key methods are utilized for assessing the failure modes of the micro-biopsy tool. The first method considers buckling, considering factors such as Young's modulus and the yield strength of the material. For a tool made of nickel, susceptibility to buckling is significant. The second method focuses on calculating the maximum transverse tip force, using variables like the distance from the neutral axis to the shank's outermost edge and the second moment of the cross-sectional area. In the case of a tool with specific dimensions and material properties, such as a 3 mm long shank length and a 67.0 µm radius of gyrus made of nickel, the tool is prone to buckling. Meanwhile, following the assumption of linear elastic deformation, the maximum force exerted by the 3 mm long micro-biopsy tool is calculated to be 0.27 N, equivalent to a bending moment of 0.81 mN-m.

(Line : 242-258).

Stress evaluations of the micro-biopsy apparatus were carried out via COMSOL Multiphysics® simulations, as depicted in Fig. 1. This simulation validated the upper tolerances for bending and buckling stress in the instrument. The design of the tool was tailored to mitigate penetration and dissection forces, eliminating the risks of shank buckling and tip fractures. For forces below 0.3 N, no buckling phenomena were observed (Fig. 2(b)(ii)). At a lateral force of 0.3 N, a peak Von Mises stress of 3.36 × 107 N/m2 was identified at the tip. Bending stress reached a maximum at the interface between the shank and the base, registering 3.49 × 107 N/m2 and 1.05 × 108 N/m2 for vertical forces of 10.0 mN and 30.0 mN, respectively (Fig. 2(b)(iii) and Fig. 2(b)(iv)).

Analytical calculations based on linear elastic deformation models suggested that the 3 mm-long tool can withstand a maximum force of 0.27 N, translating to a bending moment of 0.81 mN-m, as illustrated in Fig. 3. These data align with the simulation findings, verifying the tool's resilience to bending and buckling stress. Additional results concerning alternative designs can be found in Supplementary Fig. S1. Considering the material's higher stiffness compared to tissue, applied forces are likely to stay under the maximum stress limit of 510 MPa, especially when the tip is sharpened to a thinness of 150 µm, allowing for safe tissue penetration without tip fractures.

  1. The exact material-model is not given at the simulations. I.e., does the electroplating alter the material properties? How big a mistake is it to handle it as a homogenous/isotropic material?

Ans.: The material properties used in the simulation were based on nickel, and the material formed through electroplating differs from regular nickel. This information has been added to the simulation section, and the content has been modified as follows.

(Line : 259-266).

The stress tests and deformation models offer significant evidence for the mechanical durability of the micro-biopsy device against forces that could cause it to bend or buckle. Both the COMSOL Multiphysics® simulations and the analytical estimates confirm the device's ability to perform as expected. However, the study has limitations in accounting for variations in material properties like Young's modulus, homogeneity, and heterogeneity, specifically when comparing traditional nickel alloys to electroplated nickel. Additional studies are recommended for a more comprehensive understanding of these material differences.

  1. No detailed conclusion is drawn from the various simulated scenarios: What limitations were found? where does the model need modifications? Where will it - probably - break?

Ans.: The limitations between the simulation results and reality, as well as the maximum deformation points and possible failure points that can be reflected in the design, have been described and the content has been modified as follows.

(Line : 251-266).

Analytical calculations based on linear elastic deformation models suggested that the 3 mm-long tool can withstand a maximum force of 0.27 N, translating to a bending moment of 0.81 mN-m, as illustrated in Fig. 3. These data align with the simulation findings, verifying the tool's resilience to bending and buckling stress. Additional results concerning alternative designs can be found in Supplementary Fig. S1. Considering the material's higher stiffness compared to tissue, applied forces are likely to stay under the maximum stress limit of 510 MPa, especially when the tip is sharpened to a thinness of 150 µm, allowing for safe tissue penetration without tip fractures.

The stress tests and deformation models offer significant evidence for the mechanical durability of the micro-biopsy device against forces that could cause it to bend or buckle. Both the COMSOL Multiphysics® simulations and the analytical estimates confirm the device's ability to perform as expected. However, the study has limitations in accounting for variations in material properties like Young's modulus, homogeneity, and heterogeneity, specifically when comparing traditional nickel alloys to electroplated nickel. Additional studies are recommended for a more comprehensive understanding of these material differences.

(Line : 366-373).

Both computational simulations and laboratory experiments point to an unexpected finding that the area most susceptible to deformation in the biopsy tool is not the tip, which is often assumed to be the weak point, but rather the connection where the needle attaches to the base. This realization necessitates a shift in focus, directing attention to the structural integrity of the needle-base junction. Based on these findings, the study calls for a comprehensive redesign of the tool's 2D schematic. Specifically, the inclusion of geometric features like rounded or chamfered shapes is recommended as a strategy to enhance the tool's overall mechanical stability and durability.

  1. The paper would highly benefit from any kind of real mechanical measurement that would support the validity of the simulations. The authors shall do a more profound search into the state-of-the-art, and present that.

Ans.: Relevant latest technologies regarding the validity of the simulation have been presented and rewritten in the discussion section.

(Line : 349-355).

The specific mechanical properties can vary greatly depending on the conditions of deposition, post-treatment, and even the quality of the base material[47]. Recently, simulation methods are also being used to predict mechanical characteristics through experiments on the conditions of deposition[48] and simulations on fluid and electric field distribution[49]. Through careful control of deposition conditions and by employing advanced simulation techniques, it's possible to tailor the mechanical properties of electroplated nickel to meet these specific needs.

Comments on the Quality of English Language Numerous typos and grammatical errors can be identified.

Ans.: The entire manuscript was revised following the reviewer’s comment.

Round 2

Reviewer 2 Report

The reviewer welcomes the authors' efforts toward improving the paper. Although the reviewer maintains his opinion, that the paper could highly benefit from some further complementary simulations and real-world experiments, the aforementioned deficiencies of the study are now at least addressed in the paper. A small remark from the reviewer (regarding the new paragraph from line 366) is that in the case of cantilever beams - which is the right mechanical model of the needle for simulations - from a mechanical point of view it is not surprising at all, that the "weakest point" is where it is attached to the base, this is the expected outcome in most cases. However, the derived conclusions are sufficient!

Minor spelling and grammatical mistakes are present.